# A case-controlled study investigating gender differences in Face Name Hobby Recall (FNHR) performance in healthy community-dwelling older adults

**Luke D. Braun**[1], **H. Raymond Allen**[2], **Robbie A. Beyl**[3], **Jeffrey N. Keller**[1]*

**1** Institute for Dementia Research and Prevention, Pennington Biomedical Research Center, Baton Rouge, Louisiana, United States of America, **2** Computing Services, Population Health Intervention Information Systems, Pennington Biomedical Research Center, Baton Rouge, Louisiana, United States of America, **3** Biostatistics, Pennington Biomedical Research Center, Baton Rouge, Louisiana, United States of America

* jeffrey.keller@pbrc.edu

## Abstract

Difficulties remembering the faces and the names of individuals are two of the most common memory complaints among older adults (OA's), with impaired performance on face-name recall tests implicated to be one of the earliest changes during the transition to Alzheimer's disease. Studies in children, young-, and middle-aged adults have identified that females generally perform better on face-name association tests than males, although little is known in terms of female versus male performance in OA's. Studies in these same age groups have identified the existence of a "gender bias" whereby face-name recall is improved when facial images are from the same sex as the individual being evaluated. In the current study we employed a case-controlled study design to evaluate 115 OA males and 115 OA females in terms of their performance on the Face Name Hobby Recall (FNHR) test. OA females were observed to have significantly higher levels of both immediate and delayed recall on the FHNR test as compared to males. Improved FNHR test performance by females persisted for up to 12-months in the subset of 21 males and 21 females in the study for whom longitudinal data was available. The rates of learning for names and hobbies did not significantly differ between OA males and females. OA males and females did not exhibit improved FNHR test performance for facial images of their same sex, although OA males did show improved FHNR test performance with female faces as compared to male faces. Data from the current study have implications for future studies that examine the causes and consequences of perturbations in face-name recall in the context of aging and dementia-related research.

## Introduction

The ability of an individual to learn and remember face-name associations, and pair additional contextual information to faces, are critical components of establishing and maintaining relationships. Faces are an incredibly rich and complex source of information that

**Data availability statement:** In order to comply with legal/ethical restrictions and the policies of our institution – Pennington Biomedical Research Center – on sharing a dataset containing potentially sensitive/identifying human research participant data, we will be providing the deidentified data on request. Deidentified data can be made available for researchers who can provide evidence of IRB approval for access by contacting corresponding author Jeffrey Keller (jeffrey.keller@pbrc.edu).

**Funding:** This work was funded by funds to the Institute for Dementia Research and Prevention at the Pennington Biomedical Research Center. The funders played no role in the work or submission other than providing the funds to allow the work to be completed.

**Competing interests:** The authors have declared that no competing interests exist.

the brain interprets subconsciously and nearly instantly (milliseconds) for both facial and emotional traits [1,2]. The underlying basis by which the human brain can recognize and interpret faces has not been fully elucidated, although numerous studies have implicated the involvement of select neuroanatomical structures like the fusiform face area, superior temporal sulcus, and the anterior temporal face patch [2,3,4,5].

A meta-analysis of more than sixty peer-reviewed articles consistently found evidence of a same-gender bias for face recognition – meaning both males and females are better at learning face-name associations when facial images present as the same gender as themselves [6]. This meta-analysis focused on children/adolescents (approximate ages 6–17), young adults (approximate ages 18–35), and middle-aged adults (approximate ages 35–65) and featured only four studies of older adults (OA's) with an age greater than 65. Of these four studies, gender-bias data was only available in one. Additionally, only one study since this review has examined this topic [7]. Thus, it is not firmly established within the current literature whether OA's also exhibit a same-gender bias. In this meta-analysis and in research since, it has been established that among children, young-, and middle-aged adults, females consistently perform better than males at accurately identifying the emotional state of a person based solely on facial images [6,8]. In these same age groups females generally perform better than males in terms of their ability to learn and remember novel face-name associations [6,9–12]. Still, relatively little is known in terms of how OA females and OA males compare in terms of learning and remembering novel face-name associations [6,11,12].

A number of neurological disorders including autism spectrum disorder and "face blindness" (prosopagnosia) have been intensively investigated in the context of face-name and face-emotion studies [13,14,15]. The inability to remember face-name associations is one of the most common memory complaints reported in OA's [16,17,18]. Impairments in face-name associative test performance have been proposed to be one of earliest changes in the transition to Alzheimer's disease (AD) [19,20,21].

Taken together, these data support the continued investigation of multiple aspects of learning and recall of face-name associations in OA's. In the current study we conducted a case-controlled study of face-name-hobby association in OA's. The focus of the current study was to evaluate education and age-matched healthy community-dwelling OA males and OA females in terms of their ability to learn and recall novel face-name-hobby associations.

## Materials and methods

### Human subjects and ethical research

All data used for the current study was obtained from the web-based Louisiana Aging Brain Study (web-LABrainS). IRB approval for the web-LABrainS study was obtained from the Pennington Biomedical Research Center IRB (FWA # 00006218) prior to the initiation of web-LABrainS research efforts. The PBRC IRB approval number is 2020-044-PBRC Web-LABrainS. The study complied with ethical standards outlined in the Belmont Report and Declaration of Helsinki. Written informed consent was obtained for all participants. All participant data used in the current study was approved by all participants as part of the written informed consent. The database for the current study was created June 11, 2024. The first assessment included in the database was March 26, 2021, and the last assessment included in database was June 10, 2024.

### Web-LABrainS participants and study subset

All data in the current case-controlled study was obtained from participants in the web-based Louisiana Aging Brain Study (web-LABrainS). The web-LABrainS study is a self-guided,

web-based battery of questionnaires (demographics, mobility, medication use, health history) and assessments [22]. Web-LABrainS has been enrolling participants on a rolling basis since 2020. Participants who heard about web-LABrainS via word-of-mouth (no promotions or outreach efforts for web-LABrainS have been conducted) contacted the Pennington Biomedical Research Center (PBRC) to be enrolled. Participants were then sent an email containing a hyperlink that took them to the web-LABrainS site to consent and complete the web-LABrainS battery. Individuals were sent an email and hyperlink every 6 months to repeat the web-LABrainS battery. See Face Name Hobby Recall (FNHR) test and depressive symptom test below for further details on those assessments. Data used in the current study was selected from the larger web-LABrainS study and underwent a retrospective case-control process to match subjects based on gender. See the "Method for selecting case-controlled study subjects" section below for further details on that aspect of the study.

## Face Name Hobby Recall (FNHR) test

The self-guided FNHR test and its validation have been described previously [22]. Briefly, the FNHR test was designed to assess episodic memory and was based on the short version of the Face-Name Associative Memory Exam [21]. The eight names and hobbies were chosen based on being high frequency and short. The full list of names in the order they were presented was as follows: Linda, David, Cindy, Maria, Will, Brian, Rob, Amy. Their corresponding hobbies, also in the order they were presented, was as follows: swimming, hiking, surfing, art, baking, writing, traveling, bowling. All faces were smiling. In the learning phase, stimuli were presented at a regular 8-second cadence, and the order of facial images remained consistent throughout all trials of the FNHR assessment. The order of images in terms of race and gender was as follows: Asian female, white male, biracial female, white female, white male, white male, black male, and black female. In total, subjects completed three trials of the FNHR assessment in each session, all of which were self-administered over the course of a single session using the remote, unsupervised web-LABrainS cognitive assessment tool.

In the first two trials (T1 and T2), participants first completed the learning phase in which each face and its associated name and hobby was presented at a regular interval. Next, participants completed the test phase, in which they had to recall the associated names and hobbies when presented only with the face.

Approximately 15 minutes after the conclusion of the T2 test phase, subjects completed a third trial (T3) to test delayed recall and recognition. In this trial, subjects did not receive a learning phase and proceeded straight to the test phase.

## Other web-LABrainS measures

The components of the web-LABrainS assessment have been described previously in a study of their feasibility [22]. We will briefly summarize the components of the web-LABrainS assessments used in the current study. First, participants were presented with demographic questions about their date of birth, gender identification, racial and ethnic background, marital status, and education. Participants were then asked about their living situation, including whether they live alone and their type of residence. They were also surveyed on the types of technology they use and their self-rated comfort with technology. The next questions concerned participants' driving status and frequency. Participants were also asked how they rate their mobility and were given a short questionnaire adapted from the Life-Space Assessment [23]. Participants were then asked to provide their level of concern for their memory and whether they rate it worse than others their age. They were also asked about the number of prescription medications they use and whether these medications fall in a few common

categories. To conclude the survey portion of the web-LABrainS battery, participants were asked to provide their personal medical history by identifying the presence of current chronic health conditions.

In the next section of the web-LABrainS battery, participants took multiple cognitive assessments, including the depressive symptom assessment and the FNHR test. The psychometric validation of the depressive symptom assessment in the web-LABrainS assessment battery has been described previously [22]. The instrument was validated against pen-and-paper administration of the Geriatric Depression Scale [24] and Geriatric Anxiety Inventory [25]. The assessment contained 16 scored items developed based on widely used measures of depression and anxiety. Subjects rated the frequency/severity with which they have felt symptoms on a scale ranging from "not at all" (0) to "extremely" (4). Scores from 0–12 corresponded to subjects with low risk of depression, while scores from 13–64 corresponded to subjects with moderate to high risk of depression.

## Method for selecting case-controlled study subjects

Enrollment for the web-LABrainS study up to the point of this analysis was highly skewed toward female participants (~67% of web-LABrainS is female). This resulted in unbalanced sample sizes for our gender-based analysis of FNHR performance. Potentially confounding demographic factors like age, race, and education level also differed significantly between gender groups. With past research indicating that age and education can impact associative memory performance [18,19,26], we determined that using a case-controlled paradigm to match male and female subjects for important demographic factors and to balance sample sizes for accurate statistical analysis between gender would benefit this analysis.

To create our first case-controlled dataset, all web-LABrainS subjects with data available from a baseline visit underwent random matching for age, race, and education. First, subjects were separated into male (n=128) and female (n=381) groups. For each subject in the male group, all subjects in the female group were selected who were within two years of the male subject's age, within two years of the male subject's education level, and were of the same race as the male subject. Of the female subjects that fit those criteria, one was randomly selected to be matched to the male subject. If no such female subjects were available, the male subject was removed from the analysis. Only matched male-female pairs were included in the subsequent analysis. This resulted in an age, race, and education matched sample with male and female groups of equal sizes.

Due to the semi-randomness of the match process, different matches could result in a varying number/degree of mismatches for the parameters that were not specifically matched. To determine the matched sample to be used in the analysis, the number of mismatches for the medical conditions surveyed in the Web-LABrainS assessment (diabetes, high blood pressure, high cholesterol, thyroid deficiency, cancer, alcohol abuse, anxiety, stroke, B12 deficiency, sleep apnea, depression, concussion/traumatic brain injury, transient ischemic attack, atrial fibrillation, neurological disease, heart attack, and drug abuse) were quantified. Three matched samples were then generated, and the sample which minimized the mismatches in these medical conditions was selected for this analysis. The final dataset consisted of 115 males (13 of the original 128 were dropped for having no adequate match in the female group) and 115 females (266 of the original 381 were dropped for having no more available male subjects).

A separate sample consisting of all male (n=29) and female (n=71) subjects with data available at baseline, 6-, and 12-month visits underwent the same matching process to create our case-controlled longitudinal dataset. Once again, three matches were performed to minimize

the number of mismatches from unmatched parameters. The final longitudinal dataset consisted of 21 males (8 of the original 29 were dropped for having no adequate match in the female group) and 21 females (50 of the original 71 were dropped for having no more available male subjects).

## Statistical analyses

After applying the case-control matching protocol, male and female groups were compared for all measured variables from the larger web-LABrainS study using dependent t-tests for paired samples and chi-square tests of independence. To assess the possibility of variables that significantly differ between genders confounding our FNHR test results, we examined the relationship between the presence/absence of these variables and T1, T2, and T3 FNHR test results using mixed model ANOVA with 2×3 factorial design and Bonferroni corrected post-hoc Mann-Whitney U tests. Because these were unmatched parameters, unbalanced groups were present, necessitating the use of a non-parametric test.

Previous literature has suggested that depression and face-name associative memory are strongly associated in some age groups. To test this in our older adult sample, we computed the Spearman correlation coefficient of the relationship between depression symptom assessment scores and T1, T2, and T3 FNHR test results.

At baseline, the effect of between- and within-subject factors on FNHR performance was determined using a mixed model ANOVA with 2×3 factorial design. The total scores, name scores, and hobby scores were analyzed as separate dependent variables. For all three analyses, subject ID was treated as the between-subject identifier. Gender and trial number were treated as the dichotomous between-subject variable and three-level within-subject variable respectively. The p-values produced for the effect of gender, trial number, and the gender-trial interaction were compared to an alpha-value of 0.05. All mixed-model ANOVA p-values were Greenhouse-Geisser corrected. Post-hoc dependent t-tests for paired samples were performed between the male and female scores at each trial to determine in which trials the effect of gender produced significantly different FNHR scores. The resulting p-values underwent Bonferroni correction to reduce type I error resulting from multiple comparisons.

To determine the change in performance between the two learning trials, between the learning and memory trials, and over all trials, we calculated the difference in scores from T1 to T2, T2 to T3, and T1 to T3 for each gender group. A two-way ANOVA with 2×3 factorial design was run to determine whether the score delta significantly differed due to gender and trial gap. Once again, the total score deltas, name score deltas, and hobby score deltas were analyzed as separate dependent variables, The p-values produced for the effect of gender, trial gap, and the gender-trial gap interaction was compared to an alpha-value of 0.05. Post-hoc dependent t-tests for paired samples were performed between the male and female score deltas at each trial gap to determine at which point the effect of gender produced a significantly different improvement or decline in performance. The resulting p-values also underwent Bonferroni correction for multiple comparisons.

For our larger baseline sample, we also performed an analysis of gender bias in which we compared the average number of faces whose names were correctly identified at each trial of the (FNHR) test based on whether the face presented was of the same or different gender compared to the subject performing the assessment. To do this, we first computed a two-way repeated measures ANOVA with 2×3 factorial design. Asterisks above the brackets were then used to indicate the level of significance determined by post-hoc dependent t-tests with Bonferroni correction. A p-value was considered significant if it fell below the pre-correction alpha value threshold of 0.05. Lastly, for this sample, we also calculated and displayed the mean T1 name identification score corresponding to each face.

Longitudinal FNHR performance was analyzed using a similar methodology. Rather than examining T1, T2, and T3 FNHR performance at a single visit, FNHR performance on individual trials was examined at 0-, 6-, and 12-month visits. Mixed model ANOVA with 2×3 factorial design was again used. This time, visit served as the three-level within-subject variable. The same procedure for post-hoc testing was used to determine the visit at which significant differences in trial-specific FNHR performance occurred due to gender. We also calculated change in trial-specific FNHR performance during each gap between visits (0–6 months, 6–12 months, and 0–12 months), and ran a 2-way ANOVA with 2×3 factorial design to determine whether the trial-specific score delta significantly differed due to gender and visit gap. The same post-hoc procedure was used to determine at which visit gap the score delta significantly differed due to gender.

All statistical analysis was performed in Python [27] using functions from the numpy, scipy, and pingouin packages [28,29,30]. Outputs from the Web-LABrainS tool were stored in data structures and manipulated using objects and functions from the pandas Python package [31]. Data visualization was accomplished using the matplotlib and seaborn packages [32,33].

## Results

### Subject demographics

In this case-controlled investigation of OA males and females for FNHR test performance we analyzed a total of 230 OA's (115 males and 115 females) (Table 1). All data were obtained from participants in the web-LABrainS cohort [22], with OA male and females matched for age and education (Table 1). The mean age for OA males was 67.9 (SD 11.1) years and for females was 67.7 (SD 11.0) years (Table 1). The mean education for males was 17.1 (SD 2.4) years and for females was 17.0 (SD 2.4) years (Table 1). Study subjects were overwhelmingly white (224) and non-Hispanic (209) (Table 1).

Of the nominal data collected for this sample, OA male and female subjects differed significantly in their rates of depression, alcohol abuse, sleep apnea, thyroid deficiency, use of depression medication, use of cholesterol medication, and the rate at which they lived alone (one-way chi-square test of independence, $p < 0.05$) (Table 1). Statistical analyses revealed that only one of these variables – self-reported sleep apnea – was significantly correlated with FNHR performance during T1, T2 or T3 (Fig 1). Because depressive symptoms have been reported to modulate performance on tests involving face-name associations [34,35,36], we next conducted studies examining the correlation between the severity of depressive symptoms and performance on FNHR test (Fig 2). In these analyses, depressive symptoms in OA males and females were not significantly correlated with FNHR test scores at T1, T2, or T3 (Fig 2).

### Comparison of OA males and females on the FNHR test at baseline

OA females at baseline exhibited a significantly improved total score compared to males on T1, T2, and T3 of the FNHR assessment (Fig 3). OA females also performed significantly better on the hobby sub score of the FNHR in all three trials (Fig 3). For the name sub score of the FNHR test OA females also displayed this improved performance in T2 and T3 but not in T1 (Fig 3). At their baseline visit, both OA males and OA females tended to improve in all scores between T1 and T2 of the FNHR assessment, becoming more familiar after two exposures to the faces (Fig 3D–3F). Between T2 and T3, performance in all scores decreased as subjects underwent a delay of approximately 15 minutes from the conclusion of T2 to the start of T3 (Fig 3D–3F). OA females had a significantly smaller drop-off between T2 and T3 than

**Table 1. Demographics of study subjects sorted by gender.**

| | Total, n = 230 | Male, n = 115 | Female, n = 115 | P |
|---|---|---|---|---|
| **Age, Years, Mean (SD)** | | | | |
| Age | 67.8 (11.0) | 67.9 (11.1) | 67.7 (11.0) | 0.842 |
| **Gender, n (%)** | | | | |
| Male | 115 (50.0%) | 115 (100.0%) | 0 (0.0%) | – |
| Female | 115 (50.0%) | 0 (0.0%) | 115 (100.0%) | |
| **Race, n (%)** | | | | |
| White | 224 (97.4%) | 112 (97.4%) | 112 (97.4%) | 1 |
| Black or African American | 6 (2.6%) | 3 (2.6%) | 3 (2.6%) | |
| Asian | 0 (0.0%) | 0 (0.0%) | 0 (0.0%) | |
| American Indian or Alaska Native | 0 (0.0%) | 0 (0.0%) | 0 (0.0%) | |
| Other | 0 (0.0%) | 0 (0.0%) | 0 (0.0%) | |
| **Ethnicity, n (%)** | | | | |
| Hispanic/Latino | 1 (0.4%) | 0 (0.0%) | 1 (0.9%) | 0.109 |
| Non-Hispanic | 209 (90.9%) | 101 (87.8%) | 108 (93.9%) | |
| Other | 20 (8.7%) | 14 (12.2%) | 6 (5.2%) | |
| **Marital Status, n (%)** | | | | |
| Married | 154 (67.0%) | 86 (74.8%) | 68 (59.1%) | 0.00567 |
| Never Married | 18 (7.8%) | 11 (9.6%) | 7 (6.1%) | |
| Common-law Partner | 4 (1.7%) | 3 (2.6%) | 1 (0.9%) | |
| Divorced | 35 (15.2%) | 12 (10.4%) | 23 (20.0%) | |
| Widowed | 18 (7.8%) | 3 (2.6%) | 15 (13.0%) | |
| **Years of Education, Mean (SD)** | | | | |
| Years of Education | 17.1 (2.4) | 17.1 (2.4) | 17.0 (2.4) | 0.682 |
| **Living Situation, n (%)** | | | | |
| Living Alone | 47 (20.4%) | 15 (13.0%) | 32 (27.8%) | 0.00889 |
| Not Living Alone | 183 (79.6%) | 100 (87.0%) | 83 (72.2%) | |
| **Housing, n (%)** | | | | |
| Single Residence House | 220 (95.7%) | 107 (93.0%) | 113 (98.3%) | 0.127 |
| Assisted Living | 1 (0.4%) | 0 (0.0%) | 1 (0.9%) | |
| Apartment Complex | 4 (1.7%) | 3 (2.6%) | 1 (0.9%) | |
| Stand Alone Apartment | 3 (1.3%) | 3 (2.6%) | 0 (0.0%) | |
| Other | 2 (0.9%) | 2 (1.7%) | 0 (0.0%) | |
| **Average Technologies Used, Mean (SD)** | | | | |
| Average Technologies Used | 4.0 (1.1) | 3.9 (1.1) | 4.1 (1.0) | 0.136 |
| **Individual Technologies Used, n (%)** | | | | |
| Smartphone Use | 220 (95.7%) | 109 (94.8%) | 111 (96.5%) | 0.746 |
| Tablet Use | 179 (77.8%) | 84 (73.0%) | 95 (82.6%) | 0.112 |
| Laptop Use | 193 (83.9%) | 95 (82.6%) | 98 (85.2%) | 0.72 |
| Desktop Use | 190 (82.6%) | 97 (84.3%) | 93 (80.9%) | 0.602 |
| Wearable Use | 138 (60.0%) | 63 (54.8%) | 75 (65.2%) | 0.139 |
| **Comfort with Computers Score, Mean (SD)** | | | | |
| Comfort with Computers Score, 1–5 | 3.8 (1.5) | 3.9 (1.5) | 3.7 (1.5) | 0.244 |
| **Mobility, Mean (SD)** | | | | |
| Mobility Self-Score, 1–5 | 3.8 (0.9) | 3.9 (0.9) | 3.8 (0.9) | 0.247 |
| Life Space Mobility Score, 0–6 | 3.7 (1.2) | 3.7 (1.3) | 3.7 (1.1) | 0.912 |
| **Falls in Last Year, n (%)** | | | | |

*(Continued)*

**Table 1.** (Continued)

| | Total, n = 230 | Male, n = 115 | Female, n = 115 | P |
|---|---|---|---|---|
| Fall | 45 (19.6%) | 21 (18.3%) | 24 (20.9%) | 0.74 |
| No Fall | 185 (80.4%) | 94 (81.7%) | 91 (79.1%) | |
| **Driving Status, n (%)** | | | | |
| Regularly Drive | 223 (97.0%) | 113 (98.3%) | 110 (95.7%) | 0.157 |
| Occasionally Drive | 6 (2.6%) | 1 (0.9%) | 5 (4.3%) | |
| Rarely Drive | 0 (0.0%) | 0 (0.0%) | 0 (0.0%) | |
| Do Not Drive | 1 (0.4%) | 1 (0.9%) | 0 (0.0%) | |
| **Driving Frequency, Mean (SD)** | | | | |
| Driving Frequency Self-Score | 4.5 (0.7) | 4.6 (0.7) | 4.3 (0.7) | 0.00911 |
| **Cognition, Mean (SD)** | | | | |
| Orientation Score, 0–4 | 7.8 (0.4) | 7.8 (0.4) | 7.8 (0.4) | 0.77 |
| Immediate Recall Score, 0–4 | 4.0 (0.1) | 4.0 (0.2) | 4.0 (0.1) | 0.319 |
| Delayed Recall Score, 0–4 | 3.8 (0.6) | 3.8 (0.6) | 3.8 (0.6) | 0.915 |
| **Memory Compared to Others Their Age, n (%)** | | | | |
| Worse | 38 (16.5%) | 18 (15.7%) | 20 (17.4%) | 0.859 |
| Not Worse | 192 (83.5%) | 97 (84.3%) | 95 (82.6%) | |
| **Memory Concern Score, Mean (SD)** | | | | |
| Memory Concern Self-Score | 1.8 (0.6) | 1.9 (0.6) | 1.8 (0.6) | 0.63 |
| **Depression, Mean (SD)** | | | | |
| Depression Score, 0–64 | 12.4 (9.5) | 12.8 (9.6) | 12.0 (9.3) | 0.487 |
| **Quality of Life, Mean (SD)** | | | | |
| Quality of Life Self-Score | 80.2 (14.6) | 78.0 (14.7) | 82.4 (14.2) | 0.0207 |
| **Total Prescription Medications, Mean (SD)** | | | | |
| Total Prescription Medications | 3.5 (2.6) | 3.8 (2.7) | 3.2 (2.6) | 0.0676 |
| **Prescription Medication Types, n (%)** | | | | |
| Acid Suppression | 56 (24.3%) | 32 (27.8%) | 24 (20.9%) | 0.282 |
| Cholesterol | 116 (50.4%) | 70 (60.9%) | 46 (40.0%) | 0.00242 |
| Diabetes | 21 (9.1%) | 8 (7.0%) | 13 (11.3%) | 0.36 |
| Sleep Aids | 55 (23.9%) | 25 (21.7%) | 30 (26.1%) | 0.536 |
| Depression | 39 (17.0%) | 13 (11.3%) | 26 (22.6%) | 0.035 |
| Anxiety | 43 (18.7%) | 20 (17.4%) | 23 (20.0%) | 0.735 |
| **Medical Conditions, n (%)** | | | | |
| Diabetes | 19 (8.3%) | 6 (5.2%) | 13 (11.3%) | 0.151 |
| High Blood Pressure | 115 (50.0%) | 59 (51.3%) | 56 (48.7%) | 0.792 |
| High Cholesterol | 125 (54.3%) | 69 (60.0%) | 56 (48.7%) | 0.112 |
| Thyroid Deficiency | 33 (14.3%) | 10 (8.7%) | 23 (20.0%) | 0.024 |
| Cancer | 47 (20.4%) | 23 (20.0%) | 24 (20.9%) | 1 |
| Alcohol Abuse | 13 (5.7%) | 12 (10.4%) | 1 (0.9%) | 0.0043 |
| Anxiety | 46 (20.0%) | 21 (18.3%) | 25 (21.7%) | 0.621 |
| Stroke | 2 (0.9%) | 2 (1.7%) | 0 (0.0%) | 0.478 |
| B12 Deficiency | 11 (4.8%) | 3 (2.6%) | 8 (7.0%) | 0.216 |
| Sleep Apnea | 46 (20.0%) | 31 (27.0%) | 15 (13.0%) | 0.0134 |
| Depression | 61 (26.5%) | 20 (17.4%) | 41 (35.7%) | 0.00281 |
| Concussion or TBI | 4 (1.7%) | 2 (1.7%) | 2 (1.7%) | 1 |
| TIA | 1(0.4%) | 0(0.0%) | 1(0.9%) | 1 |
| Atrial Fibrillation | 20 (8.7%) | 14 (12.2%) | 6 (5.2%) | 0.101 |
| Neurological Disease | 7 (3.0%) | 5 (4.3%) | 2 (1.7%) | 0.443 |

*(Continued)*

**Table 1.** (Continued)

|  | Total, n = 230 | Male, n = 115 | Female, n = 115 | P |
|---|---|---|---|---|
| Heart Attack | 8 (3.5%) | 7 (6.1%) | 1 (0.9%) | 0.072 |
| Drug Abuse | 3 (1.3%) | 2 (1.7%) | 1 (0.9%) | 1 |
| Parkinson's Disease | 0 (0.0%) | 0 (0.0%) | 0 (0.0%) | 1 |

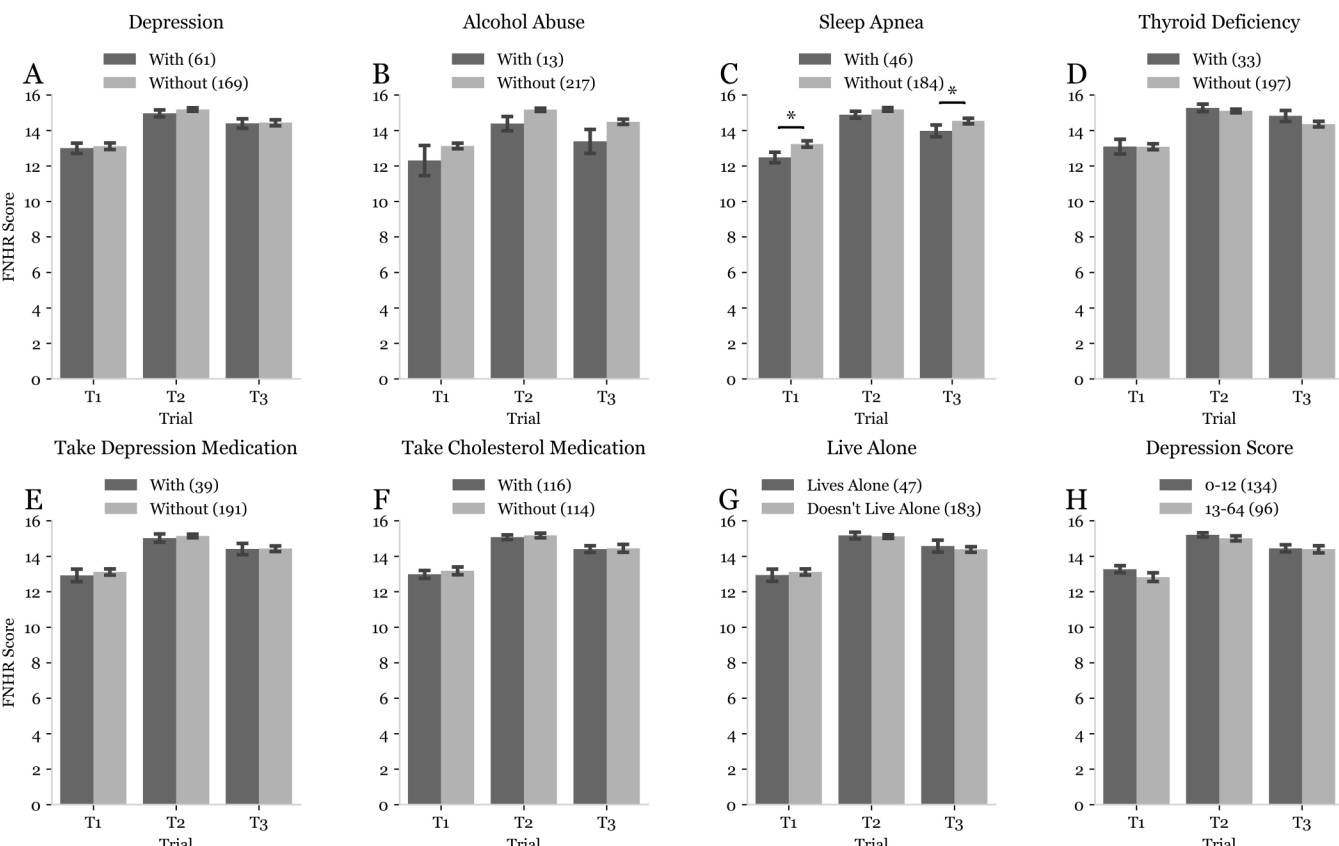

**Fig 1. Effect of demographic factors on Face Name Hobby Recall (FNHR) test performance.** Double bar plots used to depict the difference in T1, T2, and T3 Face Name Hobby Recall (FNHR) scores between subjects with (dark gray) and without (light gray) demographic parameters that were significantly different between OA males and OA females. Many of these parameters were self-reported by subjects: depression (A), alcohol abuse (B), sleep apnea (C), thyroid deficiency (D), use of depression medication (E), use of cholesterol medication (F), living alone (G). The last parameter assessed was subjects' risk of depression based on their scores on a validated depression assessment. Due to these being unmatched parameters, unbalanced groups were present. Sample sizes for each group are displayed in each figure key next to the corresponding group. Error bars were used to represent the standard error of the mean. Statistical analysis using mixed model ANOVA with 2×3 factorial design and Bonferroni corrected post-hoc Mann-Whitney U tests revealed two significant differences indicated by asterisks and brackets between the corresponding bars.

OA males for their total score and name sub score on the FNHR test (Fig 3D and 3E). Interestingly, OA males had significantly greater improvement than OA females between T1 and T2 for the hobby score on the FNHR test (Fig 3F).

We next sought to examine whether OA males and OA females differed in terms of their ability to learn and recall the names of facial images that match their sex. Neither OA males nor OA females exhibited increased performance on the FNHR test with facial images that matched the sex of the person being evaluated (Fig 4). Interestingly, we observed that OA

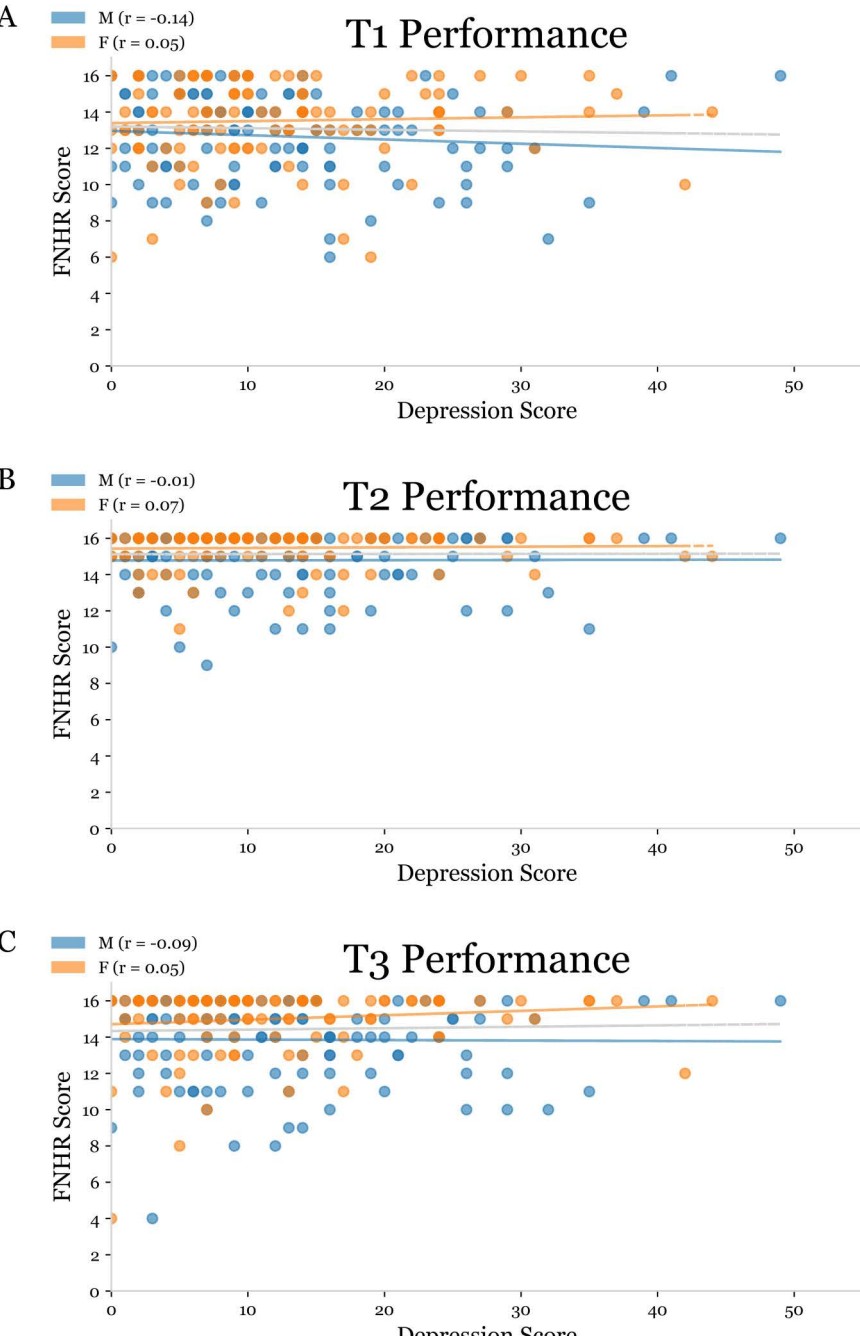

**Fig 2. Depressive symptoms and Face Name Hobby Recall (FNHR) test performance.** Scatter plots used to depict the relationship between Face Name Hobby Recall (FNHR) scores and validated depression assessment scores at T1 (A), T2 (B), and T3 (C). OA male subjects were represented by blue points, while OA females are represented by orange points. The Spearman correlation coefficient (r) for OA males and OA females was placed in the key for each plot with corresponding lines of best fit in blue, orange, and black.

males correctly identified the names of facial images for female faces significantly more often than those of male faces ([Fig 4]). The three faces for which male and female subjects had the highest level of correct name identification were a male face and two female faces (Rob, Amy, and Cindy) ([Fig 5]).

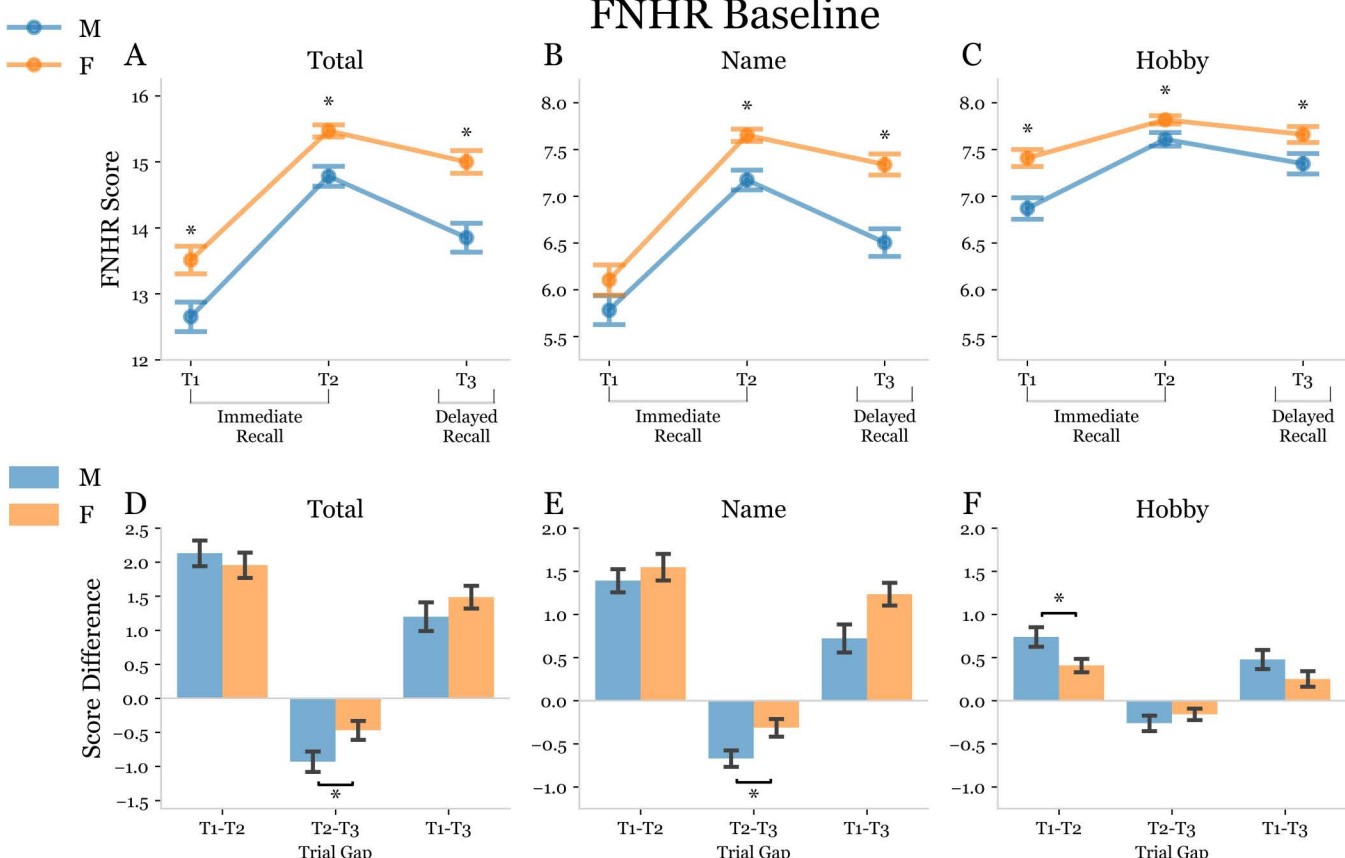

**Fig 3. Baseline Face Name Hobby Recall (FNHR) test performance.** (A-C) Split plots used to display the FNHR scores of OA males (blue) and OA females (orange) in consecutive immediate recall trials (T1 and T2) and a recall trial delayed by approximately 15 minutes (T3). Asterisks above a trial indicate a significant difference between the mean scores of male and female subjects at that trial determined using two-way ANOVA with 2×3 factorial design and subsequent post-hoc independent t-tests with Bonferroni correction for multiple comparisons. A p-value was considered significant if it fell below the pre-correction alpha value threshold of 0.05. (D-F) Double bar plots used to display the change in FNHR score from T1-T2, T2-T3, and T1-T3 for males (blue) and females (orange). A bracket between male and female bars at a given trial gap indicates a significant difference between the mean score deltas represented by the bars. An asterisk was used to indicate a significant post-hoc dependent t-test p-value following mixed model ANOVA.

## Longitudinal-based examination of FNHR test results in a subset of participants

Next, we conducted an analysis of FNHR test results in a subset of 21 males and 21 females within the case-controlled cohort for whom 6- and 12-month data was available (Table 2). OA males and OA females in this subset did not significantly differ in terms of age or education (Table 2). In our analyses we observed that the total score, name score, and hobby score for T1 were significantly higher for OA females as compared to OA males at baseline, 6- and 12-months (Fig 6). Interestingly, the total score, name score, and hobby score for T1 at 12-months was significantly increased as compared to baseline for OA females but not OA males (Fig 6). OA males did not exhibit a significant increase in total score, name score, or hobby score for T1 at 6- or 12-months (Fig 6). At T2 OA females exhibited a higher level of delayed recall for name and hobby as compared to males at baseline, 6-, and 12-months (Fig 7). The total delayed recall score (T3) for FNHR test at 12-months was significantly increased in OA females, but not OA males, at 12-months (Fig 8). OA males did not exhibit an increase in any delayed recall (T3) parameter at 6- or 12-months (Fig 8).

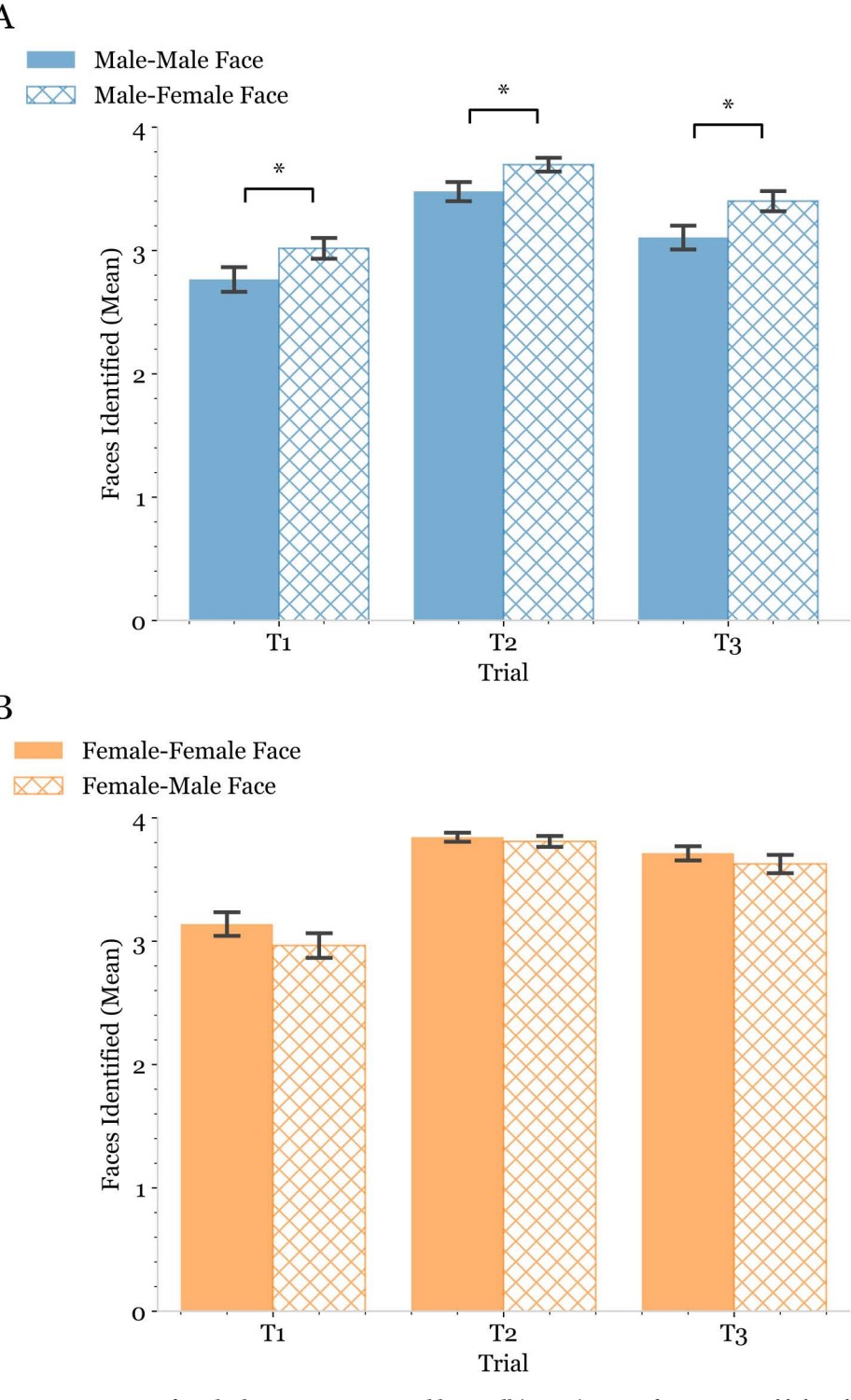

**Fig 4. Examination of gender bias on Face Name Hobby Recall (FNHR) test performance.** Double bar plot used to display the average number of faces whose names were correctly identified at each trial of the Face Name Hobby Recall (FNHR) Test based on whether the face presented was of the same gender (solid bar) or different gender (hatched bar) as the subject performing the assessment. Because there were eight faces presented in each trial and males and females were equally represented among the faces used, the maximum number of faces/names a subject could correctly identify in each trial was four. OA male subject performance was shown in blue (A), and OA female subject performance was shown in orange (B). Error bars were used to represent the standard error of the mean.

Brackets above a set of bars were used to indicate a significant difference of the means within subjects of the same gender identifying faces of different genders. Significance was determined by first computing a two-way repeated measures ANOVA with 2×3 factorial design. Asterisks above the brackets were then used to indicate the level of significance determined by post-hoc dependent t-tests with Bonferroni correction. A p-value was considered significant if it fell below the pre-correction alpha value threshold of 0.05.

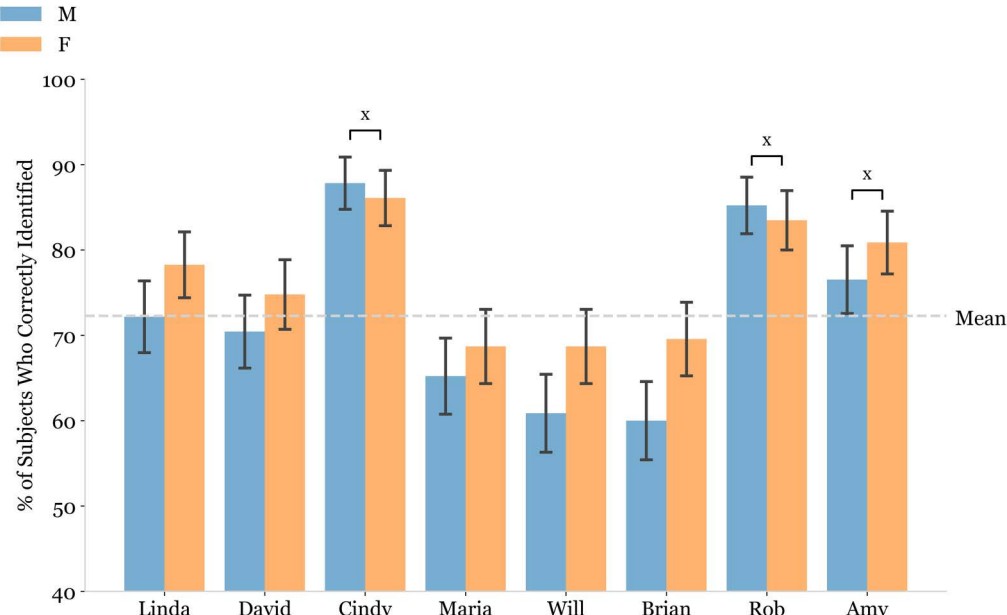

**Fig 5. Performance on specific faces that comprise the Face Name Hobby Recall (FNHR) test.** Double bar plot used to display the percentage of subjects who correctly identified the name of each face presented in Trial 1 of the FNHR test. Performance for OA male subjects was indicated in blue, while that of OA female subjects was indicated in orange. The top three faces by percentage of subjects who correctly identified them was identified for male and female subjects using an "x" symbol. If the face was in top three for both males and females, a bracket was placed between the male and female bars. Error bars were used to represent the standard error of the mean.

## Discussion

The current study adds significant new information to our current understanding of how OA males and females compare in terms of their performance on face-name association tasks. Associative memory, particularly face-name memory, is an important cognitive function for navigating the world and a strong predictor of neurodegenerative diseases that further harm mental health and well-being. Impairment of associative memory has been shown to be strongly correlated to Alzheimer's disease progression and onset of mild cognitive impairment [10,20]. Our study sheds more light on associative memory in older adults, a surprisingly understudied topic considering that this age group is most susceptible to these diseases. Forming and maintaining interpersonal relationships relies on the ability to remember the names and faces of those with whom one interacts. For this reason and others, decreased associative memory performance has also been seen in those with depression [34,35,36].

Our findings on overall face-name recall in OA's are consistent with previous studies in young and middle-aged adults demonstrating that OA females perform better on face-name association tasks as compared to OA males [8]. OA females had a higher recall of both name and hobby at every timepoint examined, as compared to OA males. Interestingly, in our longitudinal studies OA females exhibited increased recall on names and hobbies on T1

**Table 2. Demographics for participants in longitudinal study sorted by gender.**

| | Total, n = 42 | Male, n = 21 | Female, n = 21 |
|---|---|---|---|
| **Age, Years, Mean (SD)** | | | |
| Age | 68.0 (11.5) | 68.3 (11.9) | 67.6 (11.2) |
| **Gender, n (%)** | | | |
| Male | 21 (50.0%) | 21 (100.0%) | 0 (0.0%) |
| Female | 21 (50.0%) | 0 (0.0%) | 21 (100.0%) |
| **Race, n (%)** | | | |
| White | 42 (100.0%) | 21 (100.0%) | 21 (100.0%) |
| Black or African American | 0 (0.0%) | 0 (0.0%) | 0 (0.0%) |
| Asian | 0 (0.0%) | 0 (0.0%) | 0 (0.0%) |
| American Indian or Alaska Native | 0 (0.0%) | 0 (0.0%) | 0 (0.0%) |
| Other | 0 (0.0%) | 0 (0.0%) | 0 (0.0%) |
| **Ethnicity, n (%)** | | | |
| Hispanic/Latino | 1 (2.4%) | 0 (0.0%) | 1 (4.8%) |
| Non-Hispanic | 37 (88.1%) | 18 (85.7%) | 19 (90.5%) |
| Other | 4 (9.5%) | 3 (14.3%) | 1 (4.8%) |
| **Marital Status, n (%)** | | | |
| Married | 34 (81.0%) | 18 (85.7%) | 16 (76.2%) |
| Never Married | 2 (4.8%) | 2 (9.5%) | 0 (0.0%) |
| Common-law Partner | 0 (0.0%) | 0 (0.0%) | 0 (0.0%) |
| Divorced | 5 (11.9%) | 1 (4.8%) | 4 (19.0%) |
| Widowed | 1 (2.4%) | 0 (0.0%) | 1 (4.8%) |
| **Years of Education, Mean (SD)** | | | |
| Years of Education | 17.0 (2.2) | 17.1 (2.2) | 16.8 (2.3) |
| **Living Situation, n (%)** | | | |
| Living Alone | 6 (14.3%) | 1 (4.8%) | 5 (23.8%) |
| Not Living Alone | 36 (85.7%) | 20 (95.2%) | 16 (76.2%) |
| **Housing, n (%)** | | | |
| Single Residence House | 36 (85.7%) | 19 (90.5%) | 17 (81.0%) |
| Assisted Living | 0 (0.0%) | 0 (0.0%) | 0 (0.0%) |
| Apartment Complex | 1 (2.4%) | 1 (4.8%) | 0 (0.0%) |
| Stand Alone Apartment | 1 (2.4%) | 0 (0.0%) | 1 (4.8%) |
| Other | 4 (9.5%) | 1 (4.8%) | 3 (14.3%) |
| **Average Technologies Used, Mean (SD)** | | | |
| Average Technologies Used | 3.9 (0.9) | 3.8 (0.9) | 4.0 (1.0) |
| **Individual Technologies Used, n (%)** | | | |
| Smartphone Use | 40 (95.2%) | 21 (100.0%) | 19 (90.5%) |
| Tablet Use | 30 (71.4%) | 13 (61.9%) | 17 (81.0%) |
| Laptop Use | 35 (83.3%) | 14 (66.7%) | 21 (100.0%) |
| Desktop Use | 38 (90.5%) | 19 (90.5%) | 19 (90.5%) |
| Wearable Use | 22 (52.4%) | 13 (61.9%) | 9 (42.9%) |
| **Comfort with Computers Score, Mean (SD)** | | | |
| Comfort with Computers Score, 1–5 | 3.8 (1.4) | 3.6 (1.4) | 4.0 (1.4) |
| **Mobility, Mean (SD)** | | | |
| Mobility Self-Score, 1–5 | 3.9 (0.9) | 3.8 (1.0) | 4.0 (0.9) |
| Life Space Mobility Score, 0–6 | 3.5 (1.1) | 3.5 (1.2) | 3.4 (0.9) |
| **Falls in Last Year, n (%)** | | | |
| Fall | 13 (31.0%) | 8 (38.1%) | 5 (23.8%) |

*(Continued)*

**Table 2.** (Continued)

| | Total, n = 42 | Male, n = 21 | Female, n = 21 |
|---|---|---|---|
| No Fall | 29 (69.0%) | 13 (61.9%) | 16 (76.2%) |
| **Driving Status, n (%)** | | | |
| Regularly Drive | 41 (97.6%) | 21 (100.0%) | 20 (95.2%) |
| Occasionally Drive | 1 (2.4%) | 0 (0.0%) | 1 (4.8%) |
| Rarely Drive | 0 (0.0%) | 0 (0.0%) | 0 (0.0%) |
| Do Not Drive | 0 (0.0%) | 0 (0.0%) | 0 (0.0%) |
| **Driving Frequency, Mean (SD)** | | | |
| Driving Frequency Self-Score | 4.5 (0.6) | 4.7 (0.5) | 4.3 (0.6) |
| **Cognition, Mean (SD)** | | | |
| Orientation Score, 0–4 | 7.8 (0.4) | 7.9 (0.4) | 7.7 (0.5) |
| Immediate Recall Score, 0–4 | 4.0 (0.2) | 4.0 (0.2) | 4.0 (0.0) |
| Delayed Recall Score, 0–4 | 3.8 (0.7) | 3.9 (0.4) | 3.7 (0.9) |
| **Memory Compared to Others Their Age, n (%)** | | | |
| Worse | 5 (11.9%) | 4 (19.0%) | 1 (4.8%) |
| Not Worse | 37 (88.1%) | 17 (81.0%) | 20 (95.2%) |
| **Memory Concern Score, Mean (SD)** | | | |
| Memory Concern Self-Score | 1.8 (0.6) | 2.0 (0.6) | 1.6 (0.6) |
| **Depression, Mean (SD)** | | | |
| Depression Score, 0–64 | 13.1 (9.0) | 12.9 (8.1) | 13.4 (9.9) |
| **Quality of Life, Mean (SD)** | | | |
| Quality of Life Self-Score | 78.5 (14.0) | 78.3 (13.5) | 78.8 (14.9) |
| **Total Prescription Medications, Mean (SD)** | | | |
| Total Prescription Medications | 3.1 (2.4) | 3.4 (2.6) | 2.8 (2.3) |
| **Prescription Medication Types, n (%)** | | | |
| Acid Suppression | 5 (11.9%) | 3 (14.3%) | 2 (9.5%) |
| Cholesterol | 21 (50.0%) | 11 (52.4%) | 10 (47.6%) |
| Diabetes | 2 (4.8%) | 2 (9.5%) | 0 (0.0%) |
| Sleep Aids | 11 (26.2%) | 6 (28.6%) | 5 (23.8%) |
| Depression | 4 (9.5%) | 1 (4.8%) | 3 (14.3%) |
| Anxiety | 7 (16.7%) | 3 (14.3%) | 4 (19.0%) |
| **Medical Conditions, n (%)** | | | |
| Diabetes | 1 (2.4%) | 1 (4.8%) | 0 (0.0%) |
| High Blood Pressure | 19 (45.2%) | 9 (42.9%) | 10 (47.6%) |
| High Cholesterol | 20 (47.6%) | 10 (47.6%) | 10 (47.6%) |
| Thyroid Deficiency | 5 (11.9%) | 0 (0.0%) | 5 (23.8%) |
| Cancer | 7 (16.7%) | 4 (19.0%) | 3 (14.3%) |
| Alcohol Abuse | 2 (4.8%) | 2 (9.5%) | 0 (0.0%) |
| Anxiety | 9 (21.4%) | 4 (19.0%) | 5 (23.8%) |
| Stroke | 3 (7.1%) | 2 (9.5%) | 1 (4.8%) |
| B12 Deficiency | 3 (7.1%) | 1 (4.8%) | 2 (9.5%) |
| Sleep Apnea | 8 (19.0%) | 5 (23.8%) | 3 (14.3%) |
| Depression | 11 (26.2%) | 4 (19.0%) | 7 (33.3%) |
| Concussion or TBI | 1 (2.4%) | 1 (4.8%) | 0 (0.0%) |
| TIA | 1(2.4%) | 0(0.0%) | 1(4.8%) |
| Atrial Fibrillation | 4 (9.5%) | 3 (14.3%) | 1 (4.8%) |
| Neurological Disease | 2 (4.8%) | 1 (4.8%) | 1 (4.8%) |
| Heart Attack | 1 (2.4%) | 1 (4.8%) | 0 (0.0%) |

*(Continued)*

**Table 2.** (Continued)

|  | Total, n = 42 | Male, n = 21 | Female, n = 21 |
|---|---|---|---|
| Drug Abuse | 0 (0.0%) | 0 (0.0%) | 0 (0.0%) |
| Parkinson's Disease | 0 (0.0%) | 0 (0.0%) | 0 (0.0%) |

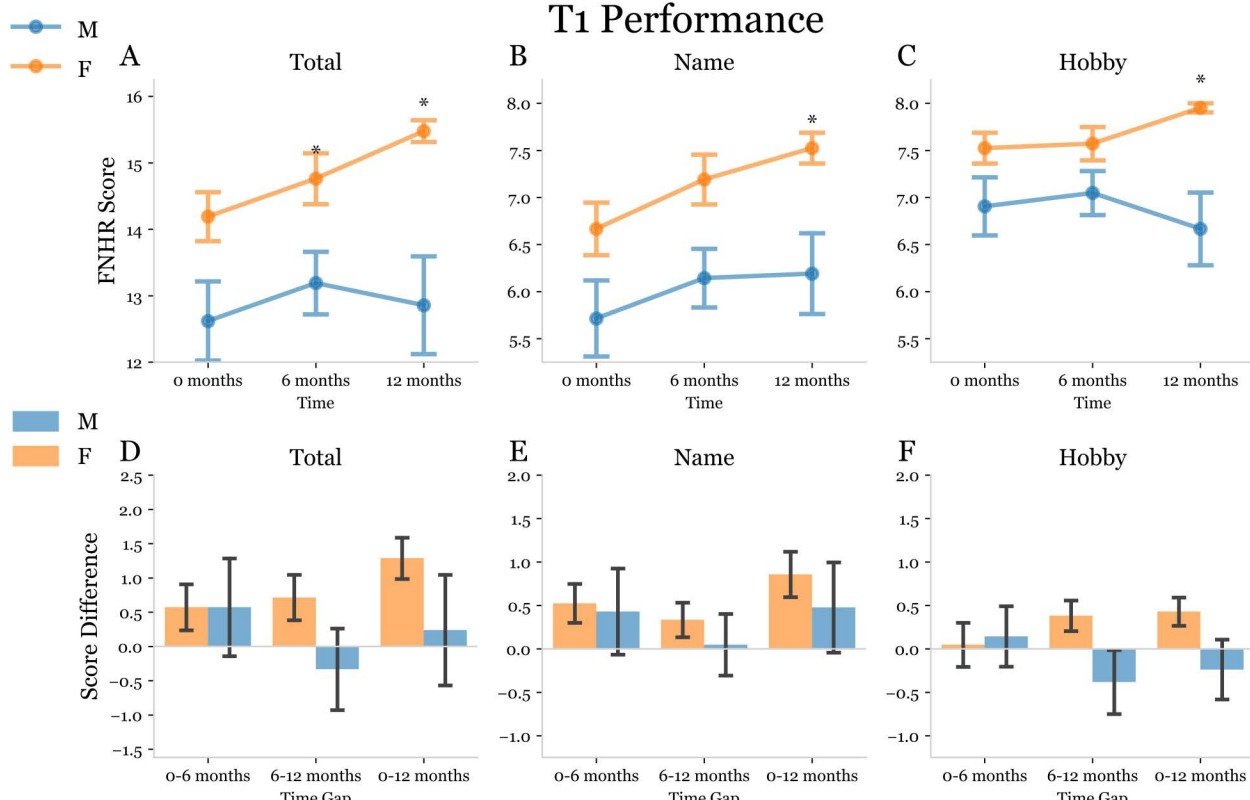

**Fig 6. Trial 1 Face Name Hobby Recall (FNHR) test performance at baseline, 6-months, and 12-months.** (A-C) Split plots used to display the Trial 1 (Immediate Recall) FNHR scores of OA males (blue) and OA females (orange) at 0-, 6-, and 12-month time points for the longitudinal subset of study subjects. Asterisks above a time point were used to indicate a significant difference between the mean scores of male and female subjects at that time point determined using two-way ANOVA with 2×3 factorial design and subsequent post-hoc dependent t-tests for paired samples with Bonferroni correction for multiple comparisons. A p-value was considered significant if it fell below the pre-correction alpha value threshold of 0.05. (D-F) Double bar plots used to display the change in FNHR score from 0–6 months, 6–12 months, and 0–12 months for males (blue) and females (orange). A bracket between male and female bars at a given time gap indicates a significant difference between the mean score deltas represented by the bars. An asterisk was used to indicate a significant post-hoc dependent t-test p-value following mixed model ANOVA.

performance at 6- and 12-months, as compared to baseline. This observation was not present in OA males. Past research has attributed the female advantage in face-name associative memory to an increased preference to attend to faces during infancy compared to males [6]. Past research also proposes a link between face-name associative memory performance and women's stronger aptitude for emotion recognition in faces [6,9]. Based on the overwhelming evidence for greater performance in OA females compared to OA males in our study, we have no reason to believe that these advantages seen earlier in life do not extend to older adulthood.

Studies of children, young-, and middle-aged adults have demonstrated evidence for increased performance on face-name tasks when the face being examined is the same sex as the person being evaluated [6,7]. In our study OA males and OA females did not perform

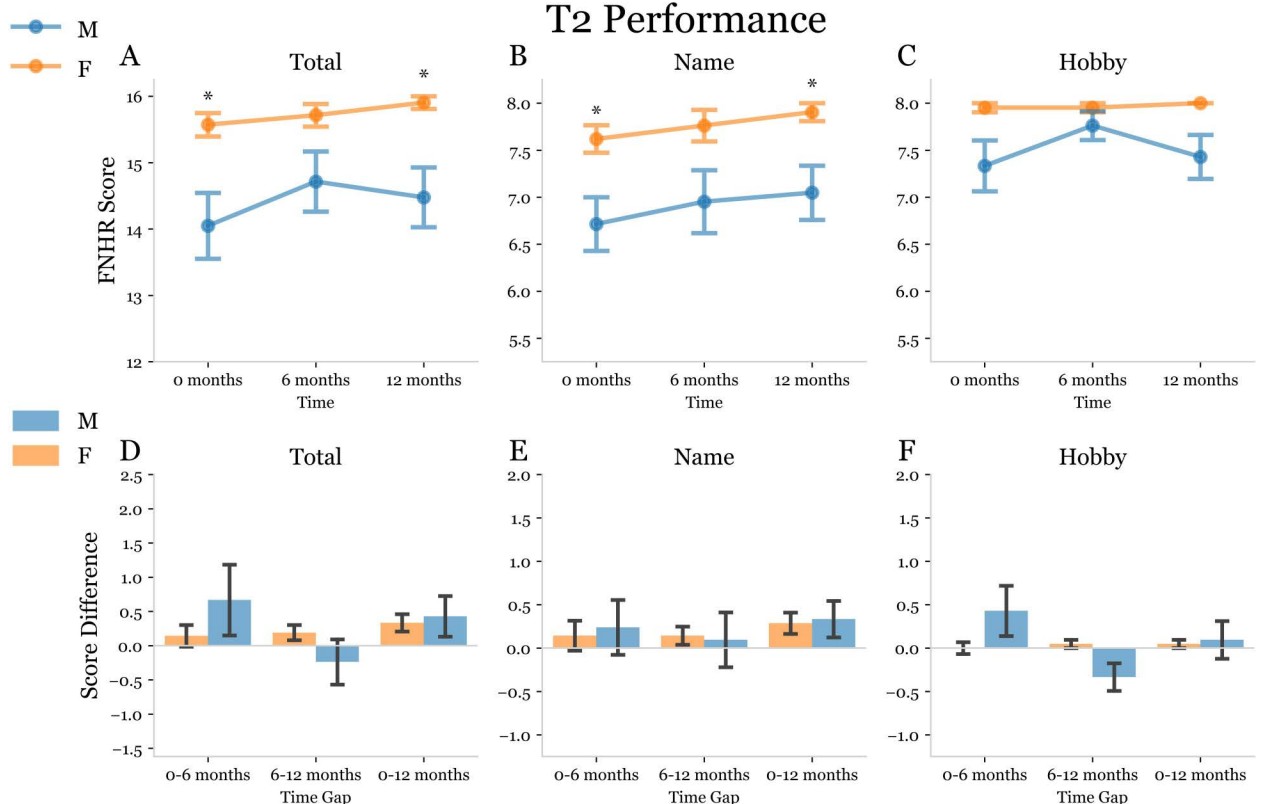

**Fig 7. Trial 2 Face Name Hobby Recall (FNHR) test performance at baseline, 6-months, and 12-months.** (A-C) Split plots used to display the Trial 1: Immediate Recall FNHR scores of OA males (blue) and OA females (orange) at 0-, 6-, and 12-month time points for the longitudinal subset of study subjects. Asterisks above a time point indicate a significant difference between the mean scores of male and female subjects at that time point determined using two-way ANOVA with 2×3 factorial design and subsequent post-hoc dependent t-tests for paired samples with Bonferroni correction for multiple comparisons. A p-value was considered significant if it fell below the pre-correction alpha value threshold of 0.05. (D-F) Double bar plots used to display the change in FNHR score from 0–6 months, 6–12 months, and 0–12 months for males (blue) and females (orange). A bracket between male and female bars at a given time gap indicates a significant difference between the mean score deltas represented by the bars. An asterisk was used to indicate a significant post-hoc dependent t-test p-value following mixed model ANOVA.

better on face-name association tests when the face examined was the same sex as the study participant. Interestingly, OA male subjects exhibited higher performance when the faces examined were female as compared to male faces. This finding diverges from gender bias results in other age groups where males rarely display a gender bias, though when they do, it tends to be a same gender bias. Women more frequently exhibit gender bias, and it also tends to be for the same gender. Past explanations for these findings have been related to in- and out-group social dynamics, in which it is proposed that individuals devote greater attentional resources to encode information about their in-group [6]. Combined with the finding that men and women get worse at facial and vocal emotion recognition in old age [9], a possible explanation for the divergent results in older adults could be less relatability to one's in-group emotionally, and therefore a smaller bias towards them in associative memory tasks.

In contrast to studies in adolescents, young-, and middle-aged adults, in the current study there was no correlation between depressive symptoms and performance on face-name association tests. Taken together, our studies suggest there are some significant differences between OA's and the findings in younger study subjects when it comes to gender bias and depression. This is particularly important given that data from younger study subjects comprise

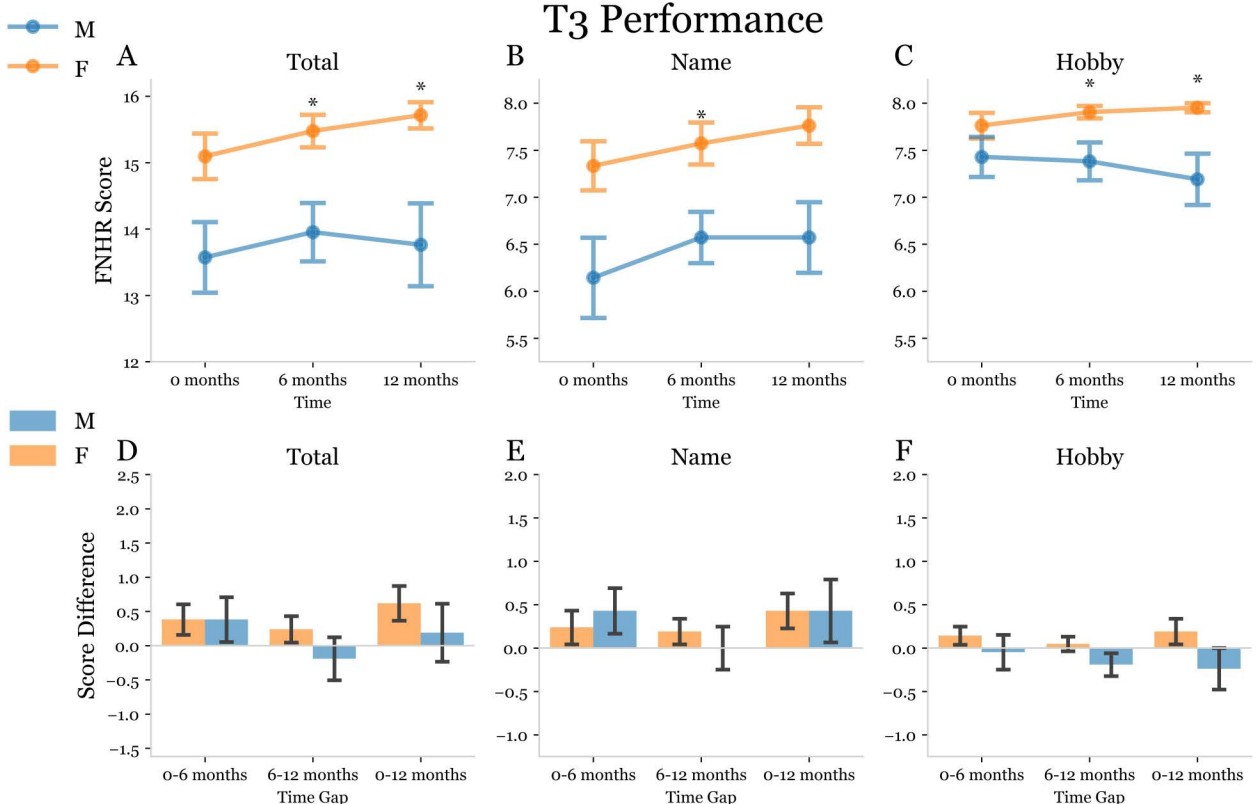

**Fig 8. Trial 3 Face Name Hobby Recall (FNHR) test performance at baseline, 6-months, and 12-months.** (A-C) Split plots used to display the Trial 3: Delayed Recall FNHR scores of OA males (blue) and OA females (orange) at 0-, 6-, and 12-month time points for the longitudinal subset of study subjects. Asterisks above a time point indicate a significant difference between the mean scores of male and female subjects at that time point determined using two-way ANOVA with 2×3 factorial design and subsequent post-hoc dependent t-tests for paired samples with Bonferroni correction for multiple comparisons. A p-value was considered significant if it fell below the pre-correction alpha value threshold of 0.05. (D-F) Double bar plots used to display the change in FNHR score from 0–6 months, 6–12 months, and 0–12 months for males (blue) and females (orange). A bracket between male and female bars at a given time gap indicates a significant difference between the mean score deltas represented by the bars. An asterisk was used to indicate a significant post-hoc dependent t-test p-value following mixed model ANOVA.

the majority of the face-name association literature. Consequently, our studies suggest that additional research is needed to clarify the full extent to which OA's overlap and differ from children, young-, and middle-aged adults on face-name association tests.

The lack of correlation between FNHR test performance and the severity of depressive symptoms in OA's was one of the more striking observations in the current study. Numerous studies in younger study subjects have reported significant correlations between depressive symptoms and face-name association test performance [34,35,36]. While additional research is needed, these findings raise the potential for face-name association learning/recall in OA to be less impacted than anticipated by mental health conditions and chronic illnesses that are well managed or of low severity. Consistent with this concept, none of the health conditions in our current study were observed to be correlated with FNHR test performance in this healthy community-dwelling sample of study subjects. Exploration of this research area may be particularly important given the growing interest in the use of face-name association tests to identify the earliest stages of AD [10,21].

Our studies support the feasibility for using web-based and self-guided assessments, including the depression measures and FNHR test, as a viable option for conducting clinical research in community dwelling OA's. In particular, our findings support the use of scalable

and low-burden web-based platforms in combination with direct recruitment efforts that incorporate social media and email-based recruitment efforts to frictionlessly enroll, consent, and assess research study subjects remotely.

The current study has a number of limitations that must be considered when interpreting the data presented and attempting to extend the current findings to the existing scientific literature. We chose to use a retrospective case-control paradigm to construct matched pairs with identical racial makeup as well as similar ages and education levels. This resulted in a more accurate gender-based comparison, but even with random matching, retrospectively constructing matches must be carefully considered during data analysis. Our case-control approach was necessary due to the current lack of targeted/balanced recruitment efforts for web-LABrainS which resulted in a study population that was overwhelmingly female, white, and highly educated. Data from this study is therefore not representative of the general population. It will be critical in future studies to attempt to replicate the current findings in a more diverse sample of study subjects. Similarly, our study sample was comprised of comparatively healthy OA's and may not reflect what occurs in OA's with uncontrolled, undiagnosed, and/or severe chronic conditions. As such, our data may play a potentially important role as a benchmark of face-name performance in OA's in future studies focused on populations with neurological, psychiatric, and/or mental health disorders. The facial images in the current FNHR study were all happy/smiling and therefore could not be used to explore the ability of OA's to identify the emotional state presented by facial images. An assessment consisting of three trials, each with eight facial images of equally represented genders, was sufficient in assessing face-name associative memory of our subjects. However, a longer assessment would provide the opportunity to represent different combinations of emotions emotional expressions, ages, races, and genders. This would also have allowed for a more accurate analysis of gender bias. Studies in OA's with a more diverse range of facial emotions, and more racial diversity of facial images, are needed in the future to increase our understanding of how face-name association in OA's compares to the large scientific literature examining children, young-, and middle-aged adults.

## Acknowledgements

The authors would like to thank all of the individuals who have participated in and continue to participate in web-LABrainS. We would also like to acknowledge Drs Qiliang He, Matthew Calamia, and Robin Hilsabeck for their help in performing a preliminary review of our work. Lastly, we would like to thank John Ruth and Aimee Stewart at PBRC for retrieving raw Face Name Hobby Recall data from the Web-LABrainS assessment tool.

## Author contributions

**Conceptualization:** Jeffrey Keller.

**Data curation:** Luke D. Braun, H. Raymond Allen.

**Formal analysis:** Luke D. Braun, Jeffrey Keller.

**Funding acquisition:** Jeffrey Keller.

**Investigation:** Jeffrey Keller.

**Methodology:** Luke D. Braun, Robbie A. Beyl, Jeffrey Keller.

**Project administration:** Jeffrey Keller.

**Resources:** Jeffrey Keller.

**Supervision:** H. Raymond Allen, Robbie A. Beyl.

**Writing – original draft:** Luke D. Braun, Jeffrey Keller.

**Writing – review & editing:** Luke D. Braun, H. Raymond Allen.

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
