## [Decision Letter · Decision Letter 0]

27 Nov 2024

PMEN-D-24-00453

A case-controlled study investigating gender differences in Face Name Hobby Recall (FNHR) performance in healthy community-dwelling older adults

PLOS Mental Health

Dear Dr. Keller,

Thank you for submitting your manuscript to PLOS Mental Health. After careful consideration, we feel that it has merit but does not fully meet PLOS Mental Health’s publication criteria as it currently stands. Therefore, we invite you to submit a revised version of the manuscript that addresses the points raised during the review process.

Be sure to:

Explicitly link your findings and outcomes to well-being. A direct link to mental health and well bring is one of our publishing criteria (https://journals.plos.org/mentalhealth/s/criteria-for-publication#loc-1). This will need to be addressed to be considered in-scope for PLOS Mental Health.

Please submit your revised manuscript by . If you will need more time than this to complete your revisions, please reply to this message or contact the journal office at mentalhealth@plos.org. Please include the following items when submitting your revised manuscript:

We look forward to receiving your revised manuscript.

Kind regards,

Jin Hui Joo

Academic Editor

PLOS Mental Health

Journal Requirements:

Additional Editor Comments (if provided):

Reviewers' comments:

Reviewer's Responses to Questions

**Comments to the Author**

1. Does this manuscript meet PLOS Mental Health’s publication criteria ? Is the manuscript technically sound, and do the data support the conclusions? The manuscript must describe methodologically and ethically rigorous research with conclusions that are appropriately drawn based on the data presented.

Reviewer #1: Yes

Reviewer #2: Yes

2. Has the statistical analysis been performed appropriately and rigorously?

Reviewer #1: Yes

Reviewer #2: Yes

3. Have the authors made all data underlying the findings in their manuscript fully available (please refer to the Data Availability Statement at the start of the manuscript PDF file)?

Reviewer #1: Yes

Reviewer #2: Yes

4. Is the manuscript presented in an intelligible fashion and written in standard English?

Reviewer #1: Yes

Reviewer #2: Yes

5. Review Comments to the Author

Reviewer #1: This study examined memory performance based on faces and names in older adults and found sex-specific differences, with female participants showing higher face-memory performance compared to male participants. The results highlight the importance of considering sex differences when investigation face-memory in older adults, and the study adds to the current literature of female adults exhibiting better memory for faces than males.

The manuscript is written nicely and contains all data. However, there are a couple of points that need to be addressed:

1) Line 48: “involvement of select neuroanatomical structures and brain networks [3,4,5].”

It would be beneficial for the reader to get some examples of involved brain structures and networks.

2) Lines 49-52 introduce the reader to the different age groups (children, young-, middle-aged, and older adults). However, the age range of each group is unknown to the reader, making it difficult to interpret previous and current findings. The authors are encouraged to further describe the age groups by age range.

3) Line 76: “with ethical standards” is written twice. Please correct.

4) The paragraph “Study participants” does not describe the participants involved in this study but rather participants involved in the main study “Web-LABrainS”. Authors are encouraged to find a different title for this paragraph.

5) In the paragraph “Face Name Hobby Recall (FNHR) test”, it is unclear if the memory task was self-administered or administered by lab personnel. More details are necessary.

6) Lines 103-105: “The order of images in terms of race and gender was as follows: Asian female, white male, biracial female, white female, white male, white male, black male, and black female.”

Why were participants shown faces of four different females in terms of race (i.e., Asian female, biracial female, White female, Black female), but only two different male faces (i.e., White male, Black male)? In order to look at memory performance of sex-specific face differences, male and female faces should be equally distributed.

7) In the paragraph “Measurement of depressive symptoms”, it is unclear which instrument was used to measure depressive symptoms. Authors should add the specific instrument and its reference for scoring to the paragraph.

8) Tables 1 and 2 show characteristics of the study sample. However, various scores are mentioned that have not been described in the methods section (i.e., Comfort with Computers, Mobility, Driving Frequency, Prescription Medications Taken, Medication Label Literacy, Quality of Life, Memory Concern), making it hard for the reader to interpret them. All characteristics listed in tables 1 and 2 should also be described in the methods section in order for the reader to be able to interpret the results.

9) Figures 1 and 2 show correlation results. However, the methods do not include correlations in the analysis section. All analytical approaches should be mentioned in the methods section.

10) Lines 317-319: “The three faces for which male and female subjects had the highest level of correct name identification were a male face and two female faces (Rob, Amy, and Cindy) (Fig 5).”

The three names do not need to be listed here since the reader does not know the full list of names used in this paradigm.

11) Figure 5 lists eight names. If those are the names used in the memory paradigm in this study, they should be included in the method section and described (e.g., common name in the study area, etc.).

12) The discussion section does not provide explanations/suggestions 1) why OA females were better in detecting faces compared to OA males, and 2) why OA males better detected female vs. male faces compared to younger male participants. The discussion would benefit from some possible interpretations and suggestions.

Reviewer #2: Describe significant physical examination and important clinical findings.

In this study, historical and current information from this episode of care was described and organised as a time line.

In order to perform the two-factor variance analysis, it is important to explain the values of the assumptions of homogeneity and normal distribution within the groups.

6. PLOS authors have the option to publish the peer review history of their article (what does this mean? ). If published, this will include your full peer review and any attached files.

**Do you want your identity to be public for this peer review?** For information about this choice, including consent withdrawal, please see our Privacy Policy .

Reviewer #1: No

Reviewer #2: **Yes: ** Sandra Paola Mondragón Bohórquez

---

## [Editor Report · Decision Letter 1]

10 Jan 2025

A case-controlled study investigating gender differences in Face Name Hobby Recall (FNHR) performance in healthy community-dwelling older adults

PMEN-D-24-00453R1

Dear Dr. Keller,

We are pleased to inform you that your manuscript 'A case-controlled study investigating gender differences in Face Name Hobby Recall (FNHR) performance in healthy community-dwelling older adults' has been provisionally accepted for publication in PLOS Mental Health.

Best regards,

Jin Hui Joo

Academic Editor

PLOS Mental Health